# Direct formation of copper nanoparticles from atoms at graphitic step edges lowers overpotential and improves selectivity of electrocatalytic CO$_2$ reduction
Tom Burwell [1], Madasamy Thangamuthu [1] ✉, Gazi N. Aliev[2], Sadegh Ghaderzadeh [1], Emerson C. Kohlrausch [1], Yifan Chen[1], Wolfgang Theis [2], Luke T. Norman[1], Jesum Alves Fernandes[1], Elena Besley [1], Pete Licence [3] & Andrei N. Khlobystov [1] ✉

A key strategy for minimizing our reliance on precious metals is to increase the fraction of surface atoms and improve the metal-support interface. In this work, we employ a solvent/ligand/counterion-free method to deposit copper in the atomic form directly onto a nanotextured surface of graphitized carbon nanofibers (GNFs). Our results demonstrate that under these conditions, copper atoms coalesce into nanoparticles securely anchored to the graphitic step edges, limiting their growth to 2–5 nm. The resultant hybrid Cu/GNF material displays high selectivity in the CO$_2$ reduction reaction (CO$_2$RR) for formate production with a faradaic efficiency of ~94% at -0.38 V vs RHE and a high turnover frequency of $2.78 \times 10^6 \, h^{-1}$. The Cu nanoparticles adhered to the graphitic step edges significantly enhance electron transfer to CO$_2$. Long-term CO$_2$RR tests coupled with atomic-scale elucidation of changes in Cu/GNF reveal nanoparticles coarsening, and a simultaneous increase in the fraction of single Cu atoms. These changes in the catalyst structure make the onset of the CO$_2$ reduction potential more negative, leading to less formate production at -0.38 V vs RHE, correlating with a less efficient competition of CO$_2$ with H$_2$O for adsorption on single Cu atoms on the graphitic surfaces, revealed by density functional theory calculations.

The rising global population and industrialisation have increased our dependence on fossil fuels to meet our energy demands, resulting in the continuous emission of carbon dioxide (CO$_2$) into the atmosphere[1]. This ongoing trend necessitates the adoption of carbon capture and utilisation (CCU) as a critical component in future carbon-neutral or low-carbon economies to mitigate environmental damage[2]. A particularly promising approach involves converting captured CO$_2$ into sustainable fuels and high-value products, as it has the potential to address both the global energy demand and the management of CO$_2$ waste into industrially important chemicals to replace the use of petrochemicals. While various methods have been explored for CO$_2$ conversion[3–8], electrocatalysis stands out as a primary choice as it offers the advantage of being compatible with renewable energy sources, allowing precise control over reaction rates and selectivity through applied voltage. Moreover, it is suitable for scaling up to industrial levels and operates efficiently under room temperature and atmospheric pressure conditions[9,10].

Numerous studies have explored the electrochemical reduction of CO$_2$ into gas products viz. CO, CH$_4$ as well as liquid products viz. formate, methanol, and ethanol, using mostly noble metal-based electrocatalysts like Pt, Au, and Pd[11–13]. These electrocatalysts are extensively studied due to their exceptional activity; however, the low abundance of these metals and high cost constrain their practical applicability. As a promising alternative, more abundant transition metals such as Cu[14], Mn[15], Co[16], Ni[17,18], and Ag[19] have been demonstrated as effective electrocatalysts for electrochemical CO$_2$ reduction. Among these, Cu stands out due to its relatively high abundance, and ability to produce alcohols, C$_2$ and C$_3$ products[14,20,21]. More importantly, specific Cu surfaces exhibit a preference for adsorbing CO$_2$ reduction intermediate carbon monoxide (CO*), over hydrogen (H*) in aqueous

[1]School of Chemistry, University of Nottingham, Nottingham, UK. [2]School of Physics & Astronomy, University of Birmingham, Birmingham, UK. [3]School of Chemistry, Carbon Neutral Laboratory, University of Nottingham, Nottingham, UK. ✉e-mail: madasamy.thangamuthu1@nottingham.ac.uk; andrei.khlobystov@nottingham.ac.uk

electrolytes, a critical factor in preventing competing water reduction[22,23]. Furthermore, the moderate CO binding energy on Cu (0.55 eV) is ideal for facilitating efficient CO adsorption and desorption, preventing electrocatalyst poisoning[20,24]. However, challenges persist when using Cu in the form of foils or large nanoparticles, as over 95% of the atoms are located below the surface and remain unutilised in the reaction[25,26]. This underscores the need of electrocatalysts in the form of single metal atoms and sub-5 nm nanoparticles to maximise atom utilization efficiency and enhance $CO_2$ reduction selectivity through the well-defined nature of catalytically active sites.

Recent advances have showcased the effectiveness of Cu single-atom catalysts (SACs) in the electrochemical reduction of $CO_2$ into $CH_4$[27,28]. For instance, Cu SACs loaded onto N-doped porous carbon have been demonstrated to efficiently generate acetone with a faradic efficiency (FE) of 36.7%. This is attributed to Cu coordination with four pyrrole-N atoms, which creates crucial active sites, lowering the $CO_2$ activation energy and promoting C-C coupling[29]. Similarly, Cu SACs decorated within an N-doped carbon matrix, offering a $CuN_4$ coordination environment, facilitate ethanol production with a 55% FE at -1.2 V vs RHE[30]. Additionally, Cu SACs deposited on carbon nanofibers selectively produce methanol with a 44% FE, involving the formation of CO* intermediate followed by its reduction[21]. Despite these successes, the stability of SACs over prolonged reaction has been a concern due to the inevitable aggregation, resulting in selectivity loss and an increase in the onset potential for $CO_2$ reduction. To address this challenge, Cu nanoclusters (CuNCs), composed of a group of atoms, have emerged as materials that combine high stability with selectivity comparable to SACs[31-33]. For instance, electrochemical $CO_2$ reduction using CuNCs at -0.75 V vs RHE produces ethylene, ethanol, and n-propanol with a collective FE of 50% and consistent activity over a 10-hour reaction, highlighting the stability of nanoclusters[34]. Oxidised CuNCs, achieved through plasma treatment, exhibit improved stability and produce ethylene with a record FE of 60%[32]. Overall, Cu-based electrocatalysts have demonstrated significant promise in $CO_2$ reduction reaction ($CO_2$RR) applications. However, precise control of the state and size distribution of active Cu centres greatly depends on the nature of the support material and the specific conditions of catalyst synthesis, which may involve wet impregnation, colloidal synthesis, or sublimation deposition methods traditionally used for the preparation of Cu nanoparticles, CuNCs or Cu SACs. In this context, engineering the metal-support interface at the atomic level and understanding its evolution during the reaction are essential to gaining precise control over $CO_2$RR electrocatalyst performance, and achieving an optimum balance of activity, selectivity, and stability.

In this study, we employ atomic deposition of Cu onto a nano-textured carbon surface to achieve a high-quality metal-carbon interface that allows investigation of electrocatalyst evolution at the atomic level, using advanced methods of electron microscopy and spectroscopy. Correlation of the structural data with the $CO_2$RR performance demonstrates that graphitic carbon step edges are of pivotal importance for the stabilisation of Cu in the form of small nanoparticles, which translates to high selectivity towards formate at low overpotentials. Augmented with computational modelling, analysis of structural changes in the Cu electrocatalyst taking place over time under $CO_2$RR conditions allows us to pinpoint the main mechanisms responsible for the loss of selectivity. For instance, the in-situ catalyst reconstruction over time shifts the $CO_2$RR onset potential resulting in changes in selectivity, thus

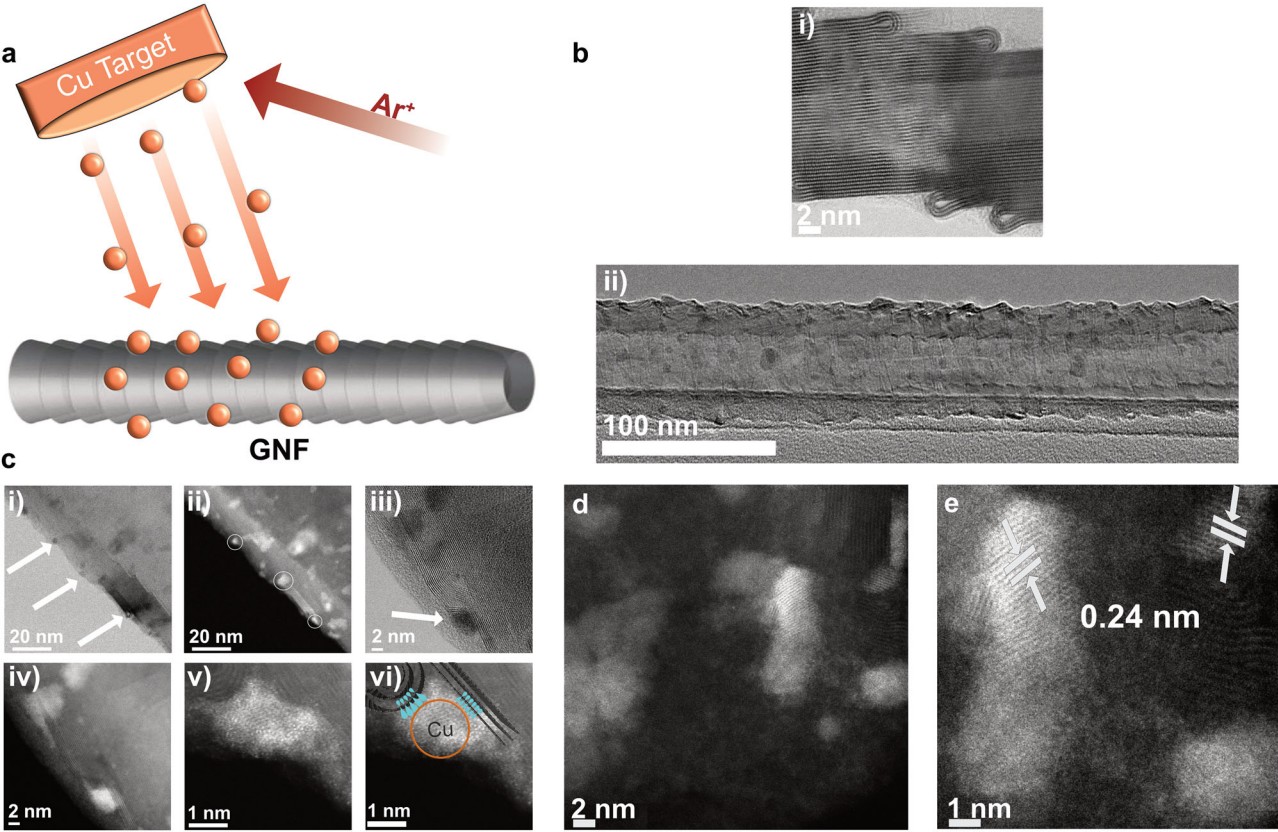

**Fig. 1 | Electrocatalyst preparation and structural characterisation. a** Schematic of magnetron sputtering delivering atoms of Cu directly onto the GNF surface, **b** (i) AC-TEM image of GNF step edges, and (ii) TEM image of Cu/GNF. **c** TEM characterisations show Cu on the step pages: (i) Bright field image with arrows indicating Cu on step edges, (ii) Dark field image with circles indicating Cu on step edges, (iii) High magnification bright field image, (iv) Dark field image, (v) High magnification image illustrating Cu wedging itself into step edge and (vi) with a very close metal-support contact, which indicates overlap of d-orbitals of the metal with π-system of the carbon lattice. **d** Low magnification and **e** magnified AC-STEM images indicating Cu species on the GNF.

**Fig. 2 | Electrochemical characterisation. a** LSV of Cu/GNF and blank GNFs measured in 0.1 M KHCO$_3$ sweeping the potential from 0.65 V to -0.85 V vs RHE with a scan rate of 10 mV s$^{-1}$. **b** Onset potential of the Cu/GNF shown for the CO$_2$RR under CO$_2$ and Ar saturated conditions, **c** Nyquist plot of GNF and Cu/GNF obtained in 0.1 M KHCO$_3$ electrolyte at a constant potential of -0.78 V vs RHE within the frequency range from 10 kHz to 0.01 Hz and **d** Tafel plot of Cu/GNF extracted from the partial current density of the CO$_2$ saturated LSV.

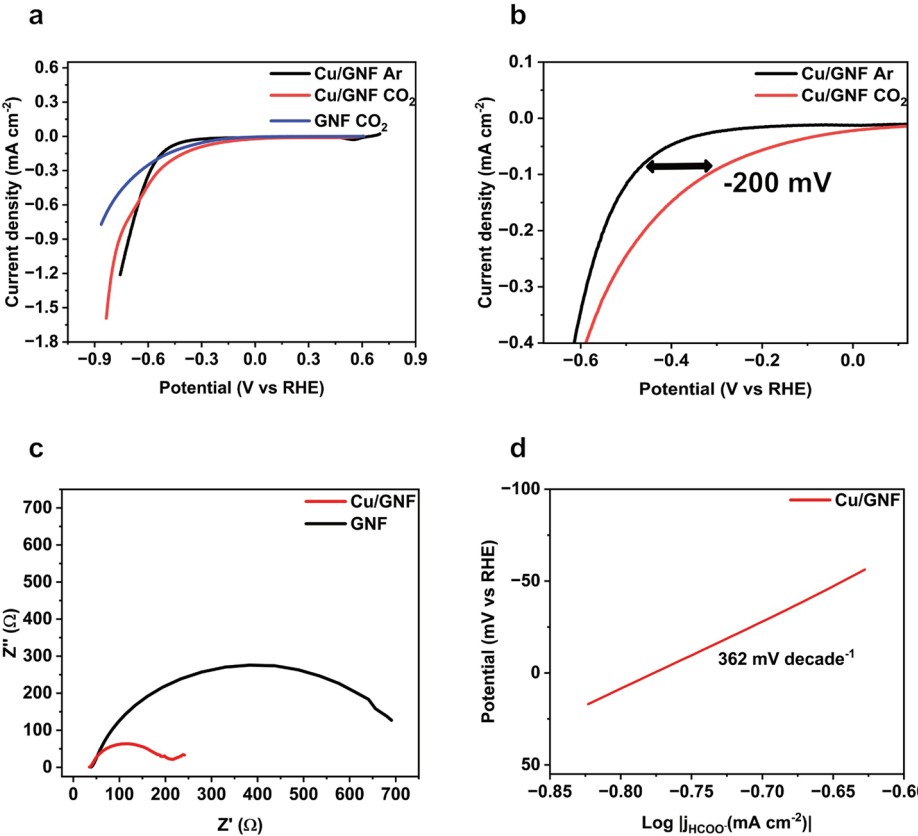

providing a strategy for the future design of highly efficient electrocatalysts for CO$_2$ reduction.

## Results and discussion
### Electrocatalyst preparation and characterisation

Magnetron sputtering was utilised for the deposition of Cu atoms directly onto GNFs (Fig. 1a), which allows for a solvent-free synthesis of metal nanoparticles with no additional impurities, such as ligands, counterions yielding pure metal in direct contact with support material produced at high rate[35]. GNFs consist of stacked graphitic cones with approximately 3 nm step edges made up of rolled-up few layers of graphene, lining the GNF surface in a direction perpendicular to the main axis (Fig. 1b). The highly textured surface of GNF presents an excellent opportunity for anchoring catalytic active centres onto the highly electrically conducting surface of GNF[36].

This approach has been exploited for improving stability[37], selectivity[38] or reusability of Pt, Pd, Rh, Cu, Au, Ru, Mo, and other metal catalysts in thermally or electrochemically activated reactions. Our recent investigations of atomistic mechanisms of this process revealed that at room temperature, metal atoms diffuse on the hexagonal lattice of the support until they become immobilised at defect sites[39], which in the case of GNFs results in the nucleation of metal nanoclusters at the graphitic step edges (Fig. 1b). Aberration-corrected scanning transmission electron microscopy (AC-STEM) imaging (Fig. 1c–e) confirms that the majority of Cu nanoparticles (NPs) are located on the step edges of the GNF (highlighted by arrows) typically reaching a diameter of 2-5 nm which appears to be dictated by the height of the step edges (Fig. 1c-iv). Based on our microscopy observations, GNF step edges (Fig. 1c-vi) can provide effective sites for Cu bonding directly to the carbon lattice, thus maximising electronic interactions between d-orbitals of the metal and π-electronic system of the graphitic layers, as evidenced by Cu "wedging" into crevices of the step edges (Fig. 1c-v), which could facilitate charge transfer between the metal and support as well as enhance the stability of the nanoparticles during reactions. Most

nanoparticles are too small to form ordered crystal-like planes of atoms, however in some cases high magnification AC-STEM images reveal patches of ordered atoms with a lattice spacing of 0.24 nm which may correspond to (111) planes in $F_{m-3m}$ phase of CuO or $P_{n-3m}$ phase of Cu$_2$O (Fig. 1e).

### Electrochemical characterisation

The electrocatalytic activity of the Cu/GNF catalyst towards CO$_2$ reduction was studied using linear sweep voltammetry (LSV). The early onset potential of -0.30 V vs RHE at 0.1 mA cm$^{-2}$ current density in the presence of CO$_2$ compared to Ar (-0.50 V vs. RHE) (Fig. 2a, b) demonstrates the activity of Cu/GNF towards CO$_2$ electrocatalytic reduction at a lower overpotential than previously reported for formate production (Eq. 1).

$$CO_2 + 2e^- + 2H^+ \rightarrow HCOOH - 0.208V\ vs\ RHE \qquad (1)$$

The cathodic sweep also highlights the major reduction peak at +0.52 V corresponding to the reduction of Cu$_2$O to Cu, and a minor peak at +0.30 V corresponding to the reduction of CuO to Cu (Figure S1)[40]. Based on these results, we propose that when Cu/GNF is exposed to air, a fraction of the metal oxidises to Cu$_2$O on the surface of GNF, which is consistent with the XPS characterisation (see later). The charge transfer resistance of the Cu/GNF was studied using electrochemical-impedance-spectroscopy (EIS) in 0.1 M KHCO$_3$ electrolyte at a constant potential of -0.78 V vs RHE within the frequency range from 10 kHz to 0.01 Hz and was used to obtain electrolyte resistance and the charge transfer resistance of the electrolyte-electrode interface.

The Nyquist plot of Cu/GNF shows a small semi-circle compared to the bare GNF suggesting that Cu loading significantly improved the charge transfer of the electrode (Fig. 2c). The solution resistance (R$_S$) is constant at 38 Ω for both electrodes, but the charge transfer resistance of the Cu/GNF is 218 Ω, significantly lower than the GNF without Cu (690 Ω) (Fig. 2c), which indicates an intimate contact between highly conducting support (GNF)

**Fig. 3 | Electrocatalytic CO$_2$ reduction. a** FE of formate obtained for the Cu/GNF under the potential ranges between -0.78 V to -0.38 V, **b** FE and the current density of Cu/GNF (0.84 wt% Cu) over time at a constant bias of -0.38 V vs RHE, **c** LSV of freshly prepared GNF, Cu/GNF and Cu/GNF after 24 h CO$_2$RR recorded with 10 mV s$^{-1}$ scan rate in 0.1 M KHCO$_3$, and **d** Zoomed version of **c** showing the onset potential of the tested catalysts. Error bars are made from three replicate measurements.

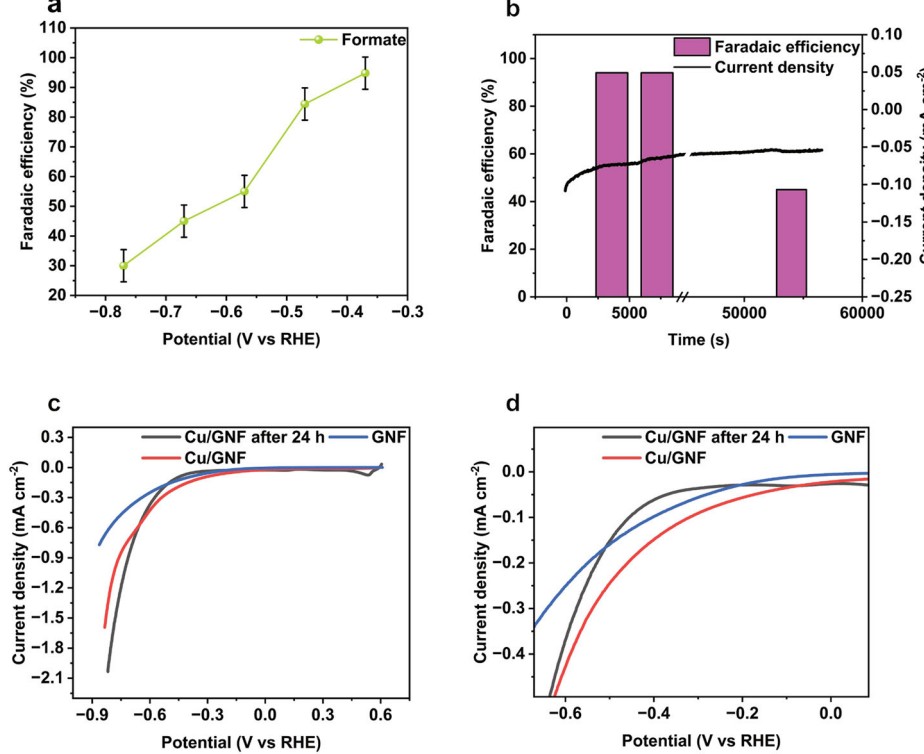

**Table 1 | Comparison of faradaic efficiencies for CO$_2$ reduction to formate for different electrocatalysts studied in this work in 0.1 M KHCO$_3$ at room temperature**

| Catalyst | Preparation method | Cu loading (wt%) | Current density at -0.38 V vs RHE (µA/cm$^2$) | FE$_{CHOO^-}$ (%) | Partial current density (µA/cm$^2$) |
|---|---|---|---|---|---|
| Cu/GNF | Atomic deposition of Cu in vacuum | 0.84 | 69 | 94 | 65 |
| Cu/GNF | Atomic deposition of Cu in vacuum | 3.38 | 200 | 18 | 36 |
| Cu/GNF | Wet chemical deposition of Cu | 0.30 | 70 | 40 | 28 |
| Cu/GNF | Wet chemical deposition of Cu | 1.32 | 90 | 60 | 54 |
| Cu foil | Commercial | N/A | 47 | 40 | 19 |
| GNF | Commercial | N/A | 3 | <1 | <<1 |

and catalytically active Cu centres. LSV was used to further explore the CO$_2$ reduction reaction mechanism by extracting the partial current density and plotting the log of current density *vs* potential (Fig. 2d). The obtained Tafel slope (Fig. 2d) value of 362 mV decade$^{-1}$ suggests that the reaction kinetics is slow and severely mass transport limited, which could be attributed to adsorbed K$^+$ blocking or limited availability of dissolved CO$_{2(aq)}$ as previously reported[41–44].

Overall, it can be concluded that the addition of Cu to GNFs greatly enhances charge transfer, while the surface of blank GNF has a large resistance and is not involved in catalysis. Therefore, Cu nanoparticles on GNFs significantly decrease charge transfer resistance thus improving the CO$_2$RR, while GNF provides a highly conducting support for Cu ensuring efficient delivery of electrons to the catalytic centres.

**Selective CO$_2$ reduction into formate**

The CO$_2$ reduction into liquid and gas products was studied using chronoamperometry at a desired constant potential in 0.1 M KHCO$_3$ in H-cell under a constant CO$_2$ concentration in solution (see experimental section for more details). $^1$H NMR spectroscopy analysis of the reaction mixture after 2 h reveals formate as the main liquid product formed by a proton-coupled reduction reaction (Eq. 1). In the range of potentials between -0.57 and -0.79 V vs RHE, faradic efficiency for formate (FE$_{CHOO^-}$) is between 55% and 30% (Fig. 3a). However, as the potential becomes less negative to

-0.38 V vs RHE, FE$_{CHOO^-}$ increases sharply to 94% with just 0.84 wt% Cu metal loading on carbon nanofibers. Surprisingly, Cu/GNF electrocatalyst with a higher content of Cu (3.38 wt%) is much less selective for formate production, with FE$_{CHOO^-}$ reaching just 18% (Figure S2). The higher Cu loadings on GNF leads to the increasing nanoparticle size (Figure S3a–c), therefore changing the properties and diminishing the formate selectivity of the Cu/GNF catalyst, which agrees well with earlier report[45].

To assess the impact of the Cu atomic deposition on GNF, we tested a similar Cu/GNF prepared by wet chemistry (Table 1). Under similar experimental conditions, wet chemistry prepared Cu/GNF electrocatalysts exhibit FE$_{CHOO^-}$ of 43% and 20% at 0.3 wt% and 1.32 wt% Cu loadings, respectively (Figure S2). Furthermore, we also tested the CO$_2$ reduction activity of Cu foil and observed FE$_{CHOO^-}$ of 40% (Figure S4). Under CO$_2$ saturation condition at a potential of -0.38 V vs RHE, faradic efficiency for H$_2$ evolution (FE$_{H_2}$) represents <10% of the overall FE, clearly demonstrating >90% selectivity to CO$_2$RR products (Fig. 3a). Whereas in the high negative potential range between -0.57 and -0.78 V vs RHE, FE$_{H_2}$ exceeded 10% (Figure S5), further corroborating the primary selectivity of Cu/GNF electrocatalysts for the CO$_2$RR at low potential(s).

Overall, the present Cu/GNF electrocatalyst demonstrated high selectivity for the formate, between -0.38 and -0.48 V vs RHE. Furthermore, the high turn-over-frequency (TOF) of 2.78 × 10$^6$ h$^{-1}$ was obtained at the lower loadings of copper on GNFs. Compared to previously reported

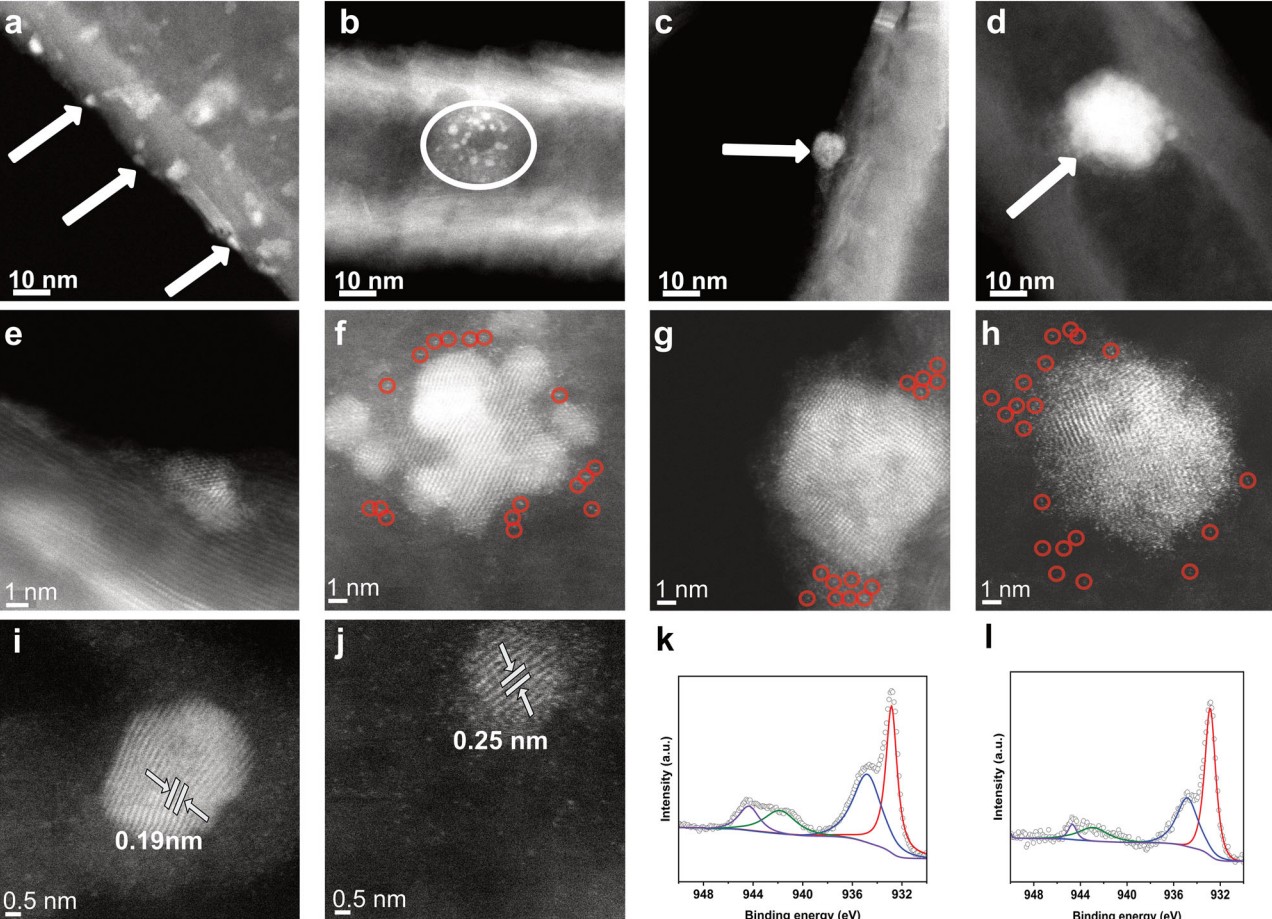

**Fig. 4 | Evolution of Cu/GNF catalyst in the CO$_2$ reduction reaction.** AC-STEM images of fresh and used Cu/GNF catalyst at -0.38 V vs RHE, **a, e** fresh, **b, f** 2 h, **c, g** 12 h, and **d, h** 24 h. Fresh Cu/GNF on step-edge illustrating no SAs, where Cu/GNF after 2 h with single atoms, and Cu/GNF after 12 h showing more single atoms. **i** Cu/GNF after 12 h and **j** Cu/GNF after 24 h showing Cu crystal structure, **k** XPS spectra of Cu/GNF before the reaction and **l** XPS spectra of Cu/GNF after 24 h reaction.

Cu-based electrocatalysts[46–48], the present Cu/GNF electrocatalyst produces formate from CO$_2$ reduction at significantly low potential (Table S3). Importantly, the metal loading method on the support plays a crucial role in the selectivity of the electrocatalyst, emphasising the significance of the quality of the metal-carbon interface for the CO$_2$RR reaction.

## Stability of the electrocatalyst

To evaluate the stability of the Cu/GNF electrocatalyst in selective formate production, an extended chronoamperometry run was performed at a constant bias of -0.38 V vs. RHE. The current density ($j$) remains practically unchanged over 24 hours (Fig. 3b) suggesting the electrocatalyst is stable under the present working condition. The selectivity for formate production at -0.38 V vs RHE remains above 90% for at least 2 h, but then gradually starts decreasing as the reaction progresses further. Analysis of the reaction solution by inductively coupled plasma optical emission spectroscopy (ICP-OES) shows no detectable leaching of Cu from Cu/GNF electrocatalyst. In a control experiment where Cu nanoparticles are supported by GNFs without external step edges but instead, with a smooth graphitic surface, exhibited low catalytic performance and hence this catalyst was not investigated in detail. This indicates that the active centres on smooth graphitic surfaces are not in a beneficial environment for CO$_2$ reduction compared to the GNF with external step edges (Figure S6a, b) and therefore show significantly less activity when compared to GNFs with step edges (Table S1). The flattening of the NP (Figure S6b) changing the surface morphology demonstrates the importance of the nanotextured surface of the support.

LSV analysis of Cu/GNF after 24 h of the CO$_2$RR shows a current density of -0.05 mA cm$^{-2}$ at -0.38 V vs RHE compared to the initial catalyst, and a negative shift of the onset potential (Fig. 3c, d). The latter must be one of the primary reasons for the drop in FE$_{CHOO^-}$ after 24 h, due to a change in the morphology of the catalyst and the emergence of single atoms (SAs) (Fig. 4f–h). It is interesting that after 24 h of the CO$_2$RR reaction, the properties of Cu/GNF prepared by atom sputtering become similar to those of Cu/GNF prepared by a wet chemistry method (Figure S7), which shows the onset potential of -0.60 V and FE$_{CHOO^-}$ of 43% right at the start of the reaction.

The cyclic voltammogram (CV) of Cu/GNF after 24 h reaction shows two major reductive and oxidative peaks at the potentials 0.52 V and 0.34 V vs RHE owing to a mixture of both Cu(I) and Cu(II) but with a more prominent Cu(II) reduction peak not seen before catalysis, confirming oxidation state changes in the electrocatalyst (Figure S8).

AC-STEM imaging (Fig. 4a–d) indicated several changes in the catalyst over time and it was observed after 2 h that SAs were present, which was not seen in imaging before the reaction, this is thought to be due to peripheral Cu breaking from larger NPs, getting stuck in defects, or detaching throughout the CO$_2$RR creating more SAs (Fig. 4e–h). Continuing from this after 12 and 24 h, an increase in NP size was observed, indicating Ostwald ripening, accompanied by the increase of SAs. This effect increases significantly from 2 to 12 h and even more so for 24 h. Image analysis at high magnification reveals a crystal lattice spacing of 0.18 nm which may correspond to (200) planes in the $F_{m-3m}$ phase of metallic Cu, and 0.25 nm which may

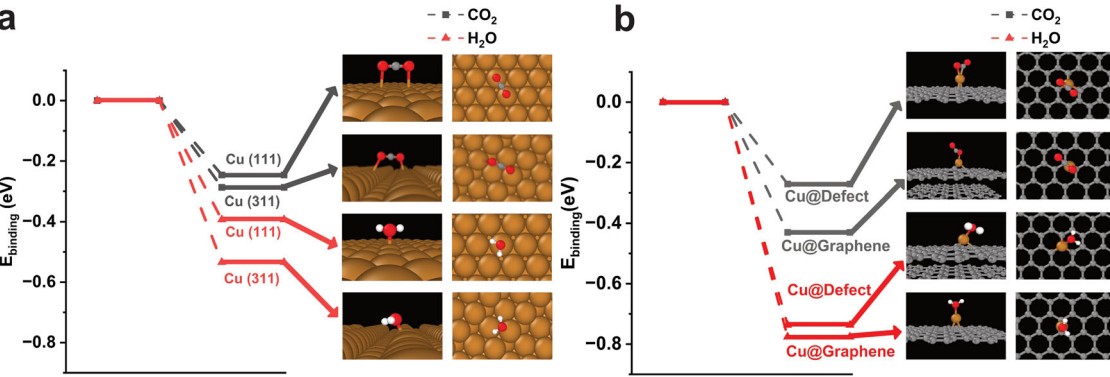

**Fig. 5 | DFT calculation.** The binding energy of $CO_2$ and $H_2O$ to **a** Cu(111) and Cu(311) surfaces of bulk metal, **b** Cu atom embedded in a vacancy defect in graphene and adsorbed on pristine graphene.

correspond to (111) planes in the $F_{m-3m}$ phase of CuO or $P_{n-3m}$ phase of $Cu_2O$ (Fig. 4i, j). The amorphous metallic copper was present before the reaction with localised domains which may be assigned to (111) planes of a copper oxide phase, but after the reaction, more SAs were present (Fig. 4f–h). Although the current density does not change significantly over the 24-h run, the FE for formate is decreased, which must be related to the changes in size and structure of Cu catalytic centres (Figure S9) (crystalline domains and SAs). The above fact can also be due to the change of Cu nanocrystalline domain and the possible phase restructuring during catalysis. For example, $CO_2$ reduction on Cu(111) surface is known to be selective for methane[14,49], due to the short residence time of other reduction products, in particular CO. In the case of other Cu phases, the residency time of the products on the surface are longer leading to more substituted products[40–42].

The oxidation states of the Cu before and after electrocatalysis were assessed using X-ray photoelectron spectroscopy (XPS). The XPS Cu 2p spectra before and after reaction, are shown in Fig. 4k, l, respectively[50]. The ratio of Cu species can be determined from the XPS Cu 2p spectra, following the methodology outlined by Biesinger and co-workers[51]. This approach assumes an overlap in the contribution of Cu(0) and Cu(I) within the Cu 2p region, making it impossible to differentiate between these two species. After 24-h reaction, the combined contribution of Cu(0) and Cu(I) increased from 33.7% to 60.4% (Table S2), suggesting that the copper species are more reduced after the electrocatalysis. Indeed, such behaviour would be expected due to the nature of the $CO_2$ reduction reaction and the applied negative potential during the reaction leading to a reduction of the copper species[52–54].

## Density functional theory calculations of $CO_2$ and $H_2O$ adsorption on copper

The selectivity of electrocatalytic reactions has been shown to be affected by the atomic surface structure of Cu face-centred lattice. For example, more stepped in nature lattice of the Cu(311) surface is selective to $CH_4$, $C_2H_4$ and $H_2$, whereas the flat lattice surface of Cu(111) is more selective to methane[55].

In order to understand the catalyst's activity changes that occurred during the $CO_2$ electrocatalysis, we studied the competitive adsorption of $H_2O$ with $CO_2$ on the Cu surface using density functional theory (DFT) calculation. It is instructive to compare the binding energies of $H_2O$ and $CO_2$ on the Cu facets of Cu(111) and Cu(311) surfaces. Our DFT results show that the binding of $H_2O$ is stronger than $CO_2$ on both surfaces (Fig. 5a), indicating that $H_2O$ reduction can compete with $CO_2$ reduction which explains the obtained ~10% FE of hydrogen evolution at -0.38 V vs RHE, in agreement with earlier reports[56,57]. The difference in the binding energy between $H_2O$ and $CO_2$ to the Cu(311) surface is 0.256 eV, which is higher than that for Cu(111), 0.147 eV, suggesting that water adsorption on Cu(311) is more favoured. As copper nanoparticles restructure during the reaction the crystal facet likely changes from Cu(111), and hence affects the

selectivity and faradic efficiency for $CO_2$ reduction products, as supported by earlier reports[23,58].

We also calculated the binding energy differences between $H_2O$ and $CO_2$ on single Cu atom adsorbed on graphene and Cu embedded in a vacancy defect site as 0.361 eV and 0.466 eV, respectively, indicating even stronger bonding of $H_2O$ (Fig. 5b) compared to the bonding on Cu nanoparticle surface. This indicates that $CO_2$RR can be weakened on Cu SAC as compared to Cu nanoparticles, thus providing a possible explanation for the observed loss of $FE_{HCOO^-}$ and the shift in the onset potential for $CO_2$RR during our long-term electrocatalysis tests when single Cu atoms start emerging on GNFs (Fig. 4f–h). Furthermore, we have performed DFT calculations for $CO_2$ and $H_2O$ adsorption on Cu SAC under external applied field of -0.38 V, to represent the similar condition to the experiment. These new results show the binding energy differences between $H_2O$ and $CO_2$ reaching 0.485 eV, 0.328 eV, and 0.367 eV while applying the electric field along X, Y, and Z-axis, respectively (Figure S10). It suggests the strong binding of $H_2O$ compared to $CO_2$ on the Cu surface, which follows the same trend as the binding energy differences obtained in the absence of an external field.

The electrochemical $CO_2$ reduction is a multi-electron and multi-proton transfer process. To date, the mechanism of $CO_2$ reduction is not fully established, but it has been reported that the mechanism of $CO_2$ reduction into formate is primarily determined by the orientation of $CO_2$ molecule adsorption on the catalyst surface, either through O atoms or C atom. Our DFT calculation shows that the $CO_2$ binds to the Cu(111) and Cu(311) surfaces via O atoms (Fig. 5a) and hence the reaction must proceed via the formation of *OCHO intermediate (Figure S11), in agreement with the currently accepted mechanisms of $CO_2$RR[59,60].

## Conclusion

Copper-on-carbon systems have been recognised among some of the most effective electrocatalysts for $CO_2$ reduction, but many nanoscale mechanisms responsible for the activity, selectivity and stability of Cu remain unanswered. In this study, we have investigated the evolution of Cu on carbon surfaces and linked nanoscale structural changes with electrocatalyst selectivity for $CO_2$RR liquid products. The mode of nanoparticle formation from Cu atoms delivered directly onto the electrically conducting support, in the absence of any solvents or reagents, ensures a detailed investigation of the metal-carbon interface during the reaction. Carbon step edges of GNF support have been shown to play a role in the initial stabilisation of Cu nanoparticles which however evolve to a mixture of larger nanoparticles and single-atoms of Cu under $CO_2$RR conditions on the timescale of 2–24 h. Metal atoms in the larger nanoparticles are more ordered than in the initial Cu nanoclusters. The larger nanoparticles possess a surface that appears to be is less attractive for $CO_2$ adsorption vs. $H_2O$ as compared to initial smaller Cu nanoparticles. The same trend holds for single Cu atoms. The structural

changes lead to the decrease of selectivity for formate production due to the onset potential shifting to more negative values, but the overall activity of Cu/GNF remains high as Cu does not desorb from the highly textured GNF surface. Importantly, the present electrocatalyst Cu/GNF exhibits very high FE for formate at low potentials, but there is a need to improve efficiency and long-term stability. As this study identified Ostwald ripening and generation of SAs on carbon surfaces to be dominant processes affecting the performance of electrocatalysts, there is a need to suppress these by designing pertinent supports to effectively stabilise Cu nanoclusters or small nanoparticles through stronger bonding to the support. As the mode of metal deposition on GNF (atomic sputtering vs wet chemistry) and metal loading both are critically important for Cu/GNF selectivity, they must be considered alongside the nature of the support for future $CO_2RR$ catalyst design.

## Methods
### Loading Cu onto GNF support
GNFs were supplied by PyroGraf (PR-24-XT-HHT) with iron content below 100 ppm. Before sputtering Cu atoms, GNFs were heat treated in air (300 °C) for 1 h to dry the surface. All depositions were carried out using an AJA magnetron sputtering system. Briefly, the GNF (0.35 g) were placed in the glove box and heated under vacuum for 5 h (100 °C) to remove any moisture. Then, the dried GNF were transferred to a custom-built stirring sample holder. The Cu deposition was carried out at room temperature with a working pressure of $3 \times 10^{-3}$ torr using Ar gas and the Cu target (99.99%). The power applied to the system was 25 W for 30 min.

### Preparation of wet chemistry Cu nanoparticles
To compare the $CO_2$ reduction activity of atomically deposited Cu on GNF electrocatalysts, we prepared Cu nanoparticles using the precipitation deposition method. In brief, 100 mg GNF was added into the 150 mL of DI water and stirred for 30 min at 80 °C. Then, 0.1 mL (for 0.34 wt %) of 11 mg/mL copper nitrate solution was added into this mixture and stirred for a further 30 min. The urea was added into the above suspension at a molar ratio of 100:1 (urea to metal) and heated at reflux for 16 h. The resultant slurry was filtered under vacuum and washed with DI water (2 L) and dried for 10 h at 110 °C. This dried catalyst is then reduced in 5% $H_2$/Ar for 1 hour at 230 °C (5 °C/min ramp rate) to remove any organic residuals[61].

### Characterisation
The amount of Cu loaded on GNFs was quantified by ICP-OES using a Perkin-Elmer Optima 2000 spectrometer, with 10 mg of the catalyst digested in aqua regia (5 mL). The morphology of the sample was studied by scanning electron microscopy (SEM) using a JEOL 7000 F Field Emission Gun microscope at 15 kV e-beam. The nanoparticle size and atomic structure were characterised by a JEOL JEM-2100F aberration-corrected scanning transmission electron microscope equipped with a Cs probe corrector (CEOS) at a convergence angle of 20 mrad and annular dark field detector (ADF) operating with an inner angle of 36 mrad and outer angle of 82 mrad at 200 kV. The bright field (BF) detector was also used in parallel. The oxidation state of the Cu was characterised by X-ray photoelectron spectroscopy (XPS) using a Kratos Axis Ultra DLD instrument, fitted with an aluminium anode, and operated at 15 kV and 10 mA with a chamber pressure of $6.7 \times 10^{-7}$ Pa. Wide energy range was acquired from 0 to 1400 eV with a step of 0.5 eV with a pass energy of 160 eV and a total scan time of 20 minutes. High-resolution scans used a step of 0.1 eV with a pass energy of 20 eV and a total scan time of 20 min. High-resolution data on the Cu 2p, O 1 s and C 1 s photoelectron peaks were collected. The X-ray source was a monochromated Al Kα emission. The energy range for each pass was calibrated using Kratos Cu $2p^{3/2}$, Ag $3d^{5/2}$ and Au $4f^{7/2}$ three-point calibration. Calibration of transmission function was performed using a clean gold sample for all lens modes and transmission generator software Vision II. The data were processed using CASAXPS and charge correction in reference to C 1 s at 284 eV.

## Electrochemical characterisation
All electrochemical experiments were performed in a standard three-electrode configuration at room temperature using the Metrohm autolab PGSTAT204 with FRAM32M module. Graphite rod and Ag/AgCl (3 M NaCl) were used as counter, and reference electrodes, respectively. The observed potentials against Ag/AgCl are iR corrected and converted into RHE using the Nernst equation: $E_{(RHE)} = E_{(Ag/AgCl)} + 0.21 + 0.0596 \times pH$. The Cu-GNF electrocatalyst thin film on carbon paper (PTFE treated (5 wt%) Toray Carbon paper-060) with a geometric surface of $1 \times 1.5\ cm^2$ was used as the working electrode. The catalyst ink was prepared by suspending the 10 mg of catalyst in 1 mL of ethanol or isopropanol and 80 μl of 5 wt% Nafion® resin followed by ultrasonication for 15 min. Then, the catalyst thin film was obtained by drop casting the 50 μl of the ink on carbon paper and dried at room temperature.

## Electrocatalysis
Electrocatalysis experiments were performed in a gas-tight two-compartment electrochemical cell (Ossila). The cathode and anode compartments were separated by Nafion®117 proton exchange membrane (Sigma Aldrich). Both compartments were filled with 30 mL of 0.1 M $KHCO_3$ solution (pH 8.34), (leaving 45 mL gas headspace) and pre-saturated with $CO_2$ for 30 min before the catalysis experiment. The carbon paper with an electrocatalyst layer and Ag/AgCl (NaCl 3 M) reference electrode was placed in the cathode compartment and the graphite rod was placed into the anode chamber. The $CO_2$ gas was continuously bubbled into the electrolyte during the reaction with a flow rate of 5 sccm. Chronoamperometry at a desired constant bias was performed with Metrohm autolab PGSTAT302N.

## Product analysis
Gas products were measured by an Agilent 8890 gas chromatography instrument equipped with a flame ionisation detector (FID) and thermal conductivity detector (TCD). High-purity Ar was used as carrier gas. The FE of the gas products was calculated using Eq. (2).

$$FE(\%) = \frac{Q_{product}}{Q_{total}} \times 100 = \frac{Z \times F \times f_{gas} \times t \times n}{Q_{total} \times 24.4x10^3} \times 100 \quad (2)$$

Where $Z$ is number of electrons to form one mole of product, $F$ is the Faraday constant, $f_{gas}$ is the flow rate of $CO_2$, $t$ is time of injection, $n$ is the number of moles of product, $24.4x10^3$ is the volume of 1 mole of gas under normal pressure and $Q_{total}$ is the charge passed at time $t$. The peak area of the product was converted to the concentration using the calibration curve, which was obtained by a standard gas mixture (see Supplementary note 1 for detailed gas product $H_2$ calculation).

The FE of the liquid products was calculated using Eq. (3).

$$FE(\%) = \frac{Q_{actual}}{Q_{total}} \times 100 = \frac{nZF}{Q_{total}} \times 100 \quad (3)$$

Where $Q_{actual}$ is the amount of charge needed to form $n$ moles of product, $Z$ is the electrons involved in the reaction and $F$ is the Faraday constant. $Q_{total}$ is the total amount of charge passed at the given time. The liquid products were measured by $^1H$ NMR spectroscopy using a Bruker AV(III) 500 with solvent ($H_2O$) suppression using Eq. (3). An aliquot of the electrolyte (400 μL) is added to $D_2O$ (48 μL) and DMSO (40 μL, 4 mM) as an internal standard, and the concentration was calculated using Eq. (4)[62].

$$C_{product} = C_{standard} \times \frac{I_{product} \times H_{Standard}}{H_{product} \times I_{standard}} \quad (4)$$

The $C_{standard}$, $I_{standard}$ and $H_{standard}$ are the concentration of the prepared standard (4 mM), the integrated area of internal standard and the number of hydrogens present on the standard, respectively. The $C_{product}$, $I_{product}$ and $H_{product}$ are the concentration of the product, the integrated area

of the product peak, and the number of hydrogens present in the product molecule, respectively. Then, the FE of liquid products were calculated using Eq. (3) (see Supplementary note 2 for detailed formate FE calculation). The turn-over frequency of the catalyst was calculated using Eq. (5) and further corrected by using Eq. (6 & 7)[63,64].

$$TOF = \frac{j_{tot} \times FE_{HCOO^-}}{2F \times n_{tot}} \quad (5)$$

Where $j_{tot}$ and $n_{tot}$ represent total current density at steady state, the number of moles of copper atoms determined by ICP-OES, $FE_{CHOO^-}$ is the FE of formate and $F$ is the Faraday constant.

$$TOF_{corrected} = \frac{TOF}{f} \quad (6)$$

Where $f$ is equal to the ratio between surface-active Cu on the working electrode from the integrated charge of the anodic wave $n$ (Figure S12) and the total moles determined via ICP-OES $n_{tot}$ (Eq. 7).

$$f = \frac{n}{n_{tot}} \times 100 \quad (7)$$

## Density functional theory calculations

Spin-polarized Density Functional Theory (DFT) calculations were performed with the Vienna Ab initio Simulation Package (VASP)[65,66] using the projector augmented-wave (PAW) method and the Perdew–Burke–Ernzerhof (PBE) exchange-correlation functional[67]. The force tolerance of $0.03$ eV$\text{Å}^{-1}$ and $0.005$ eV $\text{Å}^{-1}$, the electronic convergence of $10^{-5}$ eV and $10^{-6}$ eV and the energy cut-off of $660$ eV and $450$ eV were used for the Cu-surface and graphene, respectively. The $\Gamma$-point-centred Monkhorst−Pack $k$-point grid of $4 \times 4 \times 1$ was used to sample the Brillouin zone in both cases. Van der Waals interactions were considered using the DFT-D3 method[68], with the Becke–Johnson damping function. The Cu(111) and Cu(311) periodic slab supercells consist of five and eight layers and contain 80 and 64 Cu atoms, respectively, and the graphene supercell contains 96 C atoms. The system size and calculation setup for the Cu(111) slab were adopted from a previous study[69].

## Data availability

Any relevant data are available from the authors upon reasonable request.

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

## Acknowledgements

We acknowledge the financial support by the ESPRC/SFI CDT in Sustainable Chemistry – Atoms 2 Products (EP/S022236/1) and the EPSRC Programme Grant 'Metal Atoms on Surfaces and Interfaces (MASI) for Sustainable Future' (EP/V000055/1). E.B. acknowledges a Royal Society Wolfson Fellowship. CPU time is provided by the University of Nottingham's Augusta HPC service and the Sulis Tier 2 HPC platform funded by EPSRC Grant EP/T022108/1 and the HPC Midlands + consortium.

## Author contributions

T.B., M.T., and A.N.K. developed the methodology. E.C.K., T.B., L.T.N. and J.A.F. prepared materials. T.B. and M.T. performed electrochemical experiments. G.N.A., Y.C., and W.T. carried out electron microscopy imaging and analysis. S.G. and E.B. performed theoretical modelling. E.C.K., P.L., and J.A.F. performed spectroscopy measurements and data interpretation. A.N.K., M.T., P.L., and E.B. supervised the work. All authors contributed to writing the manuscript.

## Competing interests

The authors declare no competing interests.
