## [Peer Review File · Communications Chemistry]

Reviewers' comments:

Reviewer #1 (Remarks to the Author):

In the manuscript, the author proposed that the Cu/GNF hybrid material exhibits remarkable electrocatalytic properties in CO₂RR, displaying high selectivity for formate production with a faradaic efficiency of approximately 94%. However, the mechanisms underlying the enhancement of selectivity towards formate are not sufficiently clear. I recommend publication if the authors can provide suitable revisions addressing the comments below.

1. The current density for CO₂RR in H-cell at -0.38 V (the highest FE) is around -0.5 mA, meaning that the yield of formate production is low, especially compared with reported literatures (J. Am. Chem. Soc. 2018, 140, 8, 2880-2889, Small, 2022, 18, 1, 2105682.). Rather than increasing FE, it is more practical to enhance the overall output. I encourage the authors to add more discussion or tables to compare the conditions, FE, and yield rates with reported results.

2. The Tafel plot can analyze the rate of an electrochemical reaction as a function of the applied potential. However, in Figure 2d, the slopes of the Tafel plots for the black and red lines are the same, indicating no difference under both conditions. This suggests that adding Cu nanoparticles on the surface of GNF does not influence the reaction kinetics. Additionally, why does the gradient of 340 mV dec⁻¹ imply that the formation of CO₂^{•-} is the rate-determining step? The legends of Figure 2d should be added, and the X-axis should be labeled as $\log|j(\text{mA cm}^{-2})|$.

3. Calculating the adsorption energies of CO₂ and H₂O on single atoms and bulk/nanoparticles can only explain the difference between CO₂ and H₂O. But it can't clarify the high selectivity of prepared Cu-GFN towards formate production. For the DFT calculation, calculating the adsorption energies under applied voltages and the adsorption energies for HCOO* might be more useful for illustrating the high selectivity for formate production.

4. For the XPS results in Figures 4K and 4L, there are only peaks corresponding to Cu (I) and Cu(II), with no peak indicating Cu(0). How can their percentages be calculated? The presence of peaks for Cu (I) and Cu (II) in Figure 4K suggests that the predominant states for the prepared Cu nanoparticles are Cu₂O or CuO, rather than metallic Cu, which is contrary to the authors' declaration that Cu nanoparticles are in the dominant state. Besides, the title "Copper Atoms" might be "Copper Nanoparticles".

5. The LSV curves for blank GNF should be included, along with the corresponding discussion.

Reviewer #2 (Remarks to the Author):

The authors prepared a series of CO₂RR catalysts by depositing copper onto graphitized carbon nanotubes for formate production in pH neutral electrolyte. They showed that the catalysts are copper nanoparticles deposited at the graphitic step edges and can achieve a high formate Faraday efficiency at a low overpotential. Through a combined experimental and theoretical approach, the authors claimed that the prepared catalysts restructured and degraded during electrocatalysis. Overall, it is difficult for me to be convinced by the motivation and novelty of this study due to major concerns as listed below.

1. The performance of CO₂RR performance of the catalysts seems to be overclaimed. There is substantial number of prior research reporting high formate Faraday efficiency using catalysts such as Sn- or Bi-based materials across a wide range of overpotentials. For industrially relevant application, the desired catalysts should be selective for high current densities, which are usually driven by relatively high overpotentials. Unless the authors could show the catalysts can produce a high partial current density of formate at relatively low overpotential, it may be misleading to claim the catalysts to be "remarkable" and "exceptional".

2. The mechanism discussed here is questionable. The mechanism for formate production might not proceed via the claimed equation 2 according to findings from the prior mechanism studies. The indicated equation 1 might not be the reaction taking place in the described experiment, where water should be the hydrogen donor. The results from the DFT calculations seem to be irrelevant to the experimental observations because of the lack of (1) proposed reaction pathways for formate production or HER and (2) proper DFT models that resemble the catalyst materials prepared in the experiment.

3. The quality of the CO₂RR data analysis also requires further improvement. For example, Figure S5 shows that the FE of the products is over 100% FE at -0.37 V vs. RHE, but below 80% for potentials more negative than -0.57 V vs. RHE. In addition, the Tafel analysis in Figure 2 in the main text should use the CO₂RR partial current density instead of total current densities. Benchmarking the results against literature would be also beneficial to validate the experimental approach used in this study. Minor issues:

4. The claimed $R_s = 27$ ohm on page 7 is inconsistent with the Figure 2c, where the ohmic resistances are different between Cu/GNF and GNF.

5. Any references to support the statement of the reduction peaks for copper on Page 6?

6. The abbreviation for CO₂ reduction is inconsistent (CO₂RR and CO₂R) across the main text. "CO₂" and "H₂O" format need revision on Page 13. Some discussions such as the one on line 218 – 220 are difficult to follow and might need further clarification.

Reviewer #3 (Remarks to the Author):

The work presents the study of a composite catalyst with Cu nanoparticles on carbon nanofibers for carbon dioxide reduction reaction, CO₂RR. The catalysts are well characterized especially through a detailed HR-TEM study with atomic resolution. The observations reported in this work on the morphological changes and catalyst reconstruction after catalysis are essential for the electrocatalysis and catalysis field, which require a deeper understanding of these processes. Therefore, I suggest publication on Communications Chemistry pending minor revisions.

I found that the abstract could be confusing, and therefore I suggest clarifying the following points.

- In the description in line 14, the authors write that the catalyst is deposited in the atomic form while they also report, in line 16, that the catalyst presents 2-5 nm Cu NPs. These two points seem conflicting. Indeed, in the paper, HR-TEM images on fresh catalysts show 2-5 Cu NPs on carbon support without the presence of isolated atoms or without showing the coalescence of single atoms into NPs during sputtering (e.g. with images at different sputtering times). Could the authors make this description clearer?

- Do the authors have direct evidence of the CO₂- radical as the starting intermediate of CO₂R? The only indirect evidence is the slope in the Tafel plot. However, since it is part of the abstract, I expected that more experiments would be provided to identify this labile intermediate. Furthermore, it occurs at a very reductive potential, which is far from that of the LSV. Can the authors supplement the discussion or provide further evidence?

- The last part of the abstract seems to contradict the title. Indeed, from the title, single atoms enhance catalysis by lowering overpotential. However, in lines 23-24, single Cu atoms "disfavor" CO₂RR. Could the authors clarify this point?

Other points:

Line 115, "loss of selectivity": in the previous lines, the authors discuss that the synthesized carbon/Cu structure favors selectivity of formate at low overpotentials. However, they do not discuss the origin of such loss of selectivity, which is reasonably due to the long-term reaction. Could the

authors clarify this point?

Lines 158-162: use of RHE and SHE; I suggest using only one reference across the MS.

Lines 202-205: the sentence is unclear and it should be rephrased.

Figure S5: the total FE is below 80% at large negative voltages. Have the authors possible explanations for the missing FE?

Line 238, "CO₂R": in the text it is mostly occurring as "CO₂RR" could the authors use only one of them?

Line 261: "(0.2 mA cm⁻² at -0.38 V vs RHE)" * probably a missing *-* due to a typo. Furthermore, the value disagrees with Figure 3D because at -0.38 V Cu/GNF after 24 h, i shows ca. -0.05 mA cm⁻².

Lines 264-267: have the authors checked with HR-TEM the morphology of the best-performing sample after CO₂RR at large negative potentials and compared it with the sample after stability tests? Perhaps this comparison could provide information on the performance loss at a high reaction rate.

Lines 294-295: "XPS of fresh catalysts I) and after 24 hours J)" this part of the caption is probably a typo.

Lines 297-301: there is probably a typo because line 301 mentions the presence of Cu(0), but table S2 shows only Cu(I) and Cu(II) - which is reasonable because Cu(0) and Cu(I) have overlapping XPS signals. A detailed study on Cu XPS signals can be found in DOI: 10.1002/sia.6239. Additionally, the presence of Cu(II) should have two signature satellite peaks at higher binding energies (ca. 934 and 944 eV) with respect to Cu(I)/Cu(0) peaks (at ca. 932 and 941 eV). Could the authors double-check the assignments?

Lines 318 and 319: CO₂ and H₂O - subscripts are missing.

Reviewer 1:

1. The current density for CO₂RR in H-cell at -0.38 V (the highest FE) is around -0.5 mA, meaning that the yield of formate production is low, especially compared with reported literatures (J. Am. Chem. Soc. 2018, 140, 8, 2880-2889, Small, 2022, 18, 1, 2105682.). Rather than increasing FE, it is more practical to enhance the overall output. I encourage the authors to add more discussion or tables to compare the conditions, FE, and yield rates with reported results.

Response: We thank the reviewer for this comment. Though the electrocatalyst activity can be characterised by the current density, and overpotential, the faradaic efficiency (FE) and the turnover frequency (TOF) are the recommended parameters to report and compare with the literature. Hence, we have compared the FE of the present work with the literature utilising the Cu-based electrocatalysts in **Table S3**, as recommended by the reviewer. The corresponding text is included in the revised main manuscript in page number 9. The reviewer highlighted literature studies have used non-Cu catalysts such as B-doped Pd catalyst (J. Am. Chem. Soc. 2018, 140, 8, 2880-2889), and Bi₂S₃-Bi₂O₃ nanosheets (Small, 2022, 18, 1, 2105682), and hence we have not included in the comparison Table S3.

Table S3. Comparison of CO₂ reduction activity of the present Cu/GNF catalyst with the Cu-based electrocatalysts reported in the literature.

Catalyst	Year	Electrolyte	Potential (V vs RHE)	FE of formate (%)	Turnover frequency (TOF)	Reference
Cu/GNF	2024	0.1 M KHCO ₃	-0.38	94	2.78×10 ⁶ h ⁻¹	This work
Cu/N-Doped porous Carbon	2023	0.1 M KHCO ₃	-0.70	52	-	1
Cu/CuO _x /SnO _x on porous carbon	2023	0.5 M KHCO ₃	-1.1	69	-	2
Cu ₁ Bi ₂ Aerogel	2022	0.5 M KHCO ₃	-0.90	96	-	3
Cu-FTGDE	2024	0.5 M KHCO ₃	-0.90	76	-	4
Cu ₂ SnS ₃	2023	0.1 M KHCO ₃	-1.20	92	-	5
SU-101-Cu@2.5C	2023	0.5 M KHCO ₃	-0.96	95	-	6
Cu/Bi ₂ S ₃ -2.67%-N ₂	2023	0.5 M KHCO ₃	-0.80	94	-	7
Pd ₇₃ Cu ₂₇	2023	0.5 M KHCO ₃	-0.56	81	-	8
Cu-Pd/MXene	2023	0.1 M KHCO ₃	-0.50	79	-	9
Bi ₉ Cu ₁	2023	0.5 M KHCO ₃	-0.80	98	-	10

2. The Tafel plot can analyse the rate of an electrochemical reaction as a function of the applied potential. However, in Figure 2d, the slopes of the Tafel plots for the black and red lines are the same, indicating no difference under both conditions. This suggests that adding Cu nanoparticles on the surface of GNF does not influence the reaction kinetics. Additionally, why does the gradient of 340 mV dec⁻¹ imply that the formation of CO₂⁻ is the rate-determining step? The legends of Figure 2d should be added, and the X-axis should be labelled as log*j* (mA cm⁻²).

Response: We thank the reviewer for this constructive comment. We admit that the previous version of **Figure 2D** might mislead the readers' understanding. Actually, **Figure 2D** represents a Tafel plot of just Cu/GNF catalyst only. Now, we have provided the revised Figure 2D. The obtained Tafel slope value of 362 mV dec^{-1} suggests that the kinetics of the present CO_2 reduction on Cu/GNF catalyst is slow and the limited diffusion/mass transport of CO_2 must be the rate-determining step, in agreement with earlier reports.^{43–46} This discussion is added in the revised main manuscript on page number 7.

Figure 2. D) Tafel plot of Cu/GNF extracted from the partial current density of the CO_2 saturated LSV.

3. Calculating the adsorption energies of CO_2 and H_2O on single atoms and bulk/nanoparticles can only explain the difference between CO_2 and H_2O . But it can't clarify the high selectivity of prepared Cu-GFN towards formate production. For the DFT calculation, calculating the adsorption energies under applied voltages and the adsorption energies for HCOO^* might be more useful for illustrating the high selectivity for formate production.

Response: We thank the reviewer for this constructive comment. We wanted to understand the catalyst's activity changes during electrocatalysis by studying the competitive adsorption of H_2O with CO_2 on the Cu surface using DFT calculation, not to explain the formate selectivity. As suggested by the reviewer, we have performed new DFT calculations for CO_2 and H_2O adsorption on the catalyst surface under an externally applied field of -0.38 V for Cu@Graphene, to represent a similar condition to the experiment. These new results show the binding energy differences between H_2O and CO_2 reaching 0.485 eV , 0.328 eV , and 0.367 eV while applying the electric field along X, Y, and Z-axis, respectively (Figure S10). It suggests the strong binding of H_2O compared to CO_2 on the Cu surface, which follows the same trend as the binding energy differences obtained in the absence of an external field. These new results are discussed in the main manuscript on page 14 and figures included in the supporting information. We considered new calculations for the samples Cu(111) and Cu(311), but as those calculations are based on a metal surface, applying an electric field to a metal causes displacement of conduction electrons which cancel out the applied potential. However, as previously stated, the trend observed in an applied electric field is the same with both CO_2 and H_2O , therefore providing good evidence that this type of modelling is a convenient way to

predict the competition of H₂O and CO₂ on a specific catalytic surface that is changing over time.

Figure S10. The binding energy of CO₂ and H₂O on Cu atom adsorbed on graphene under external applied field of -0.38 V along the X, Y and Z-axis.

4. For the XPS results in Figures 4K and 4L, there are only peaks corresponding to Cu (I) and Cu(II), with no peak indicating Cu(0). How can their percentages be calculated? The presence of peaks for Cu (I) and Cu(II) in Figure 4K suggests that the predominant states for the prepared Cu nanoparticles are Cu₂O or CuO, rather than metallic Cu, which is contrary to the authors' declaration that Cu nanoparticles are in the dominant state. Besides, the title “Copper Atoms” might be “Copper Nanoparticles”.

Response: The Cu species ratio can be determined by the XPS Cu 2p spectra, following the methodology outlined by Biesinger and co-workers (M. C. Biesinger, et al., Appl. Surf. Sci., 2010, 257, 887-898). This method presumes an overlap of the Cu⁰ and Cu⁺¹ contribution on the Cu 2p region. To better understand this, we modified the text in the revised main manuscript on page number 12, as given below, as well as the table S2 on the support information.

Revised text in the main manuscript

“The ratio of Cu species ratio can be determined from the XPS Cu 2p spectra, following the methodology outlined by Biesinger and co-workers.⁵³ This approach assumes an overlap in the contribution of Cu(0) and Cu(I) within the Cu 2p region, making it impossible to differentiate between these two species. After 24-hour reaction, the combined contribution of Cu(0) and

Cu(I) increased from 33.7% to 60.4% (**Table S2**), suggesting that the copper species are more reduced after the electrocatalysis.”

As recommended by the reviewer, we have now changed the title into “Copper Nanoparticles”.

5. The LSV curves for blank GNF should be included, along with the corresponding discussion.

Response: We thank the reviewer for this constructive comment. As recommended by the reviewer, the LSV curve for blank GNF is added to the Figure 2A in the revised version.

Reviewer 2:

1. The performance of CO₂RR performance of the catalysts seems to be overclaimed. There is substantial number of prior research reporting high formate Faraday efficiency using catalysts such as Sn- or Bi-based materials across a wide range of overpotentials. For industrially relevant application, the desired catalysts should be selective for high current densities, which are usually driven by relatively high overpotentials. Unless the authors could show the catalysts can produce a high partial current density of formate at relatively low overpotential, it may be misleading to claim the catalysts to be “remarkable” and “exceptional”.

Response: We understand the reviewer's concern about the electrochemical CO₂ reduction activity of the present Cu/GNF catalyst for formate production. Indeed, the partial current density, and faradaic efficiency of the present Cu/GNF are not comparable to the reported Sn, and Bi-based catalysts. However, the obtained high selectivity for formate with a ~94% faradaic efficiency at a low overpotential of 0.17 V vs RHE and a high turnover frequency of $2.78 \times 10^6 \text{ h}^{-1}$ is comparable to the earlier reports utilising Cu-based catalysts for CO₂ reduction. We have included a new literature comparison, Table S3, in the revised supporting information and the corresponding discussion in the main manuscript on page number 9.

Moreover, we would like to mention that this work aims to study the fundamental changes of the atomically deposited Cu catalyst over the reaction time on the GNF surface rather than competing with the benchmark activity reported in the literature. To avoid misleading the readers, we have modified the words “remarkable” and “exceptional” in the revised version of the manuscript.

2. The mechanism discussed here is questionable. The mechanism for formate production might not proceed via the claimed equation 2 according to findings from the prior mechanism studies. The indicated equation 1 might not be the reaction taking place in the described experiment, where water should be the hydrogen donor. The results from the DFT calculations seem to be irrelevant to the experimental observations because of the lack of (1) proposed reaction pathways for formate production or HER and (2) proper DFT models that resemble the catalyst materials prepared in the experiment.

Response: We thank the reviewer for this constructive comment. The electrochemical CO₂ reduction is a multi-electron and multi-proton transfer process. To date, the mechanism of CO₂ reduction is not entirely well established. Still, it has been reported that the mechanism of CO₂ reduction into formate is primarily determined by the orientation of CO₂ molecule adsorption on the catalyst surface, either through O atoms or C atoms. Our DFT calculation shows that the CO₂ binds to the Cu surface via O atoms (**Figure 5A**) and hence the reaction must proceed via formation of *OCHO intermediate (**Figure S11**), in agreement with literature.^{61,62} This revised discussion is included in the revised manuscript on page number 14.

Figure S11. Proposed mechanism for the electrochemical CO₂ reduction into formate using the present Cu/GNF catalyst.

We agree with the reviewer that the water is the proton donor in equation 1, which is commonly represented as below.

3. The quality of the CO₂RR data analysis also requires further improvement. For example, Figure S5 shows that the FE of the products is over 100% FE at -0.37 V vs. RHE, but below 80% for potentials more negative than -0.57 V vs. RHE. In addition, the Tafel analysis in Figure 2 in the main text should use the CO₂RR partial current density instead of total current densities. Benchmarking the results against literature would be also beneficial to validate the experimental approach used in this study.

Response: We thank the reviewer for these valuable comments. We have now revisited the data and plotted a revised figure S5 (below). The results show that the FE is slightly over 100% within error levels originating from the NMR measurements and the gas chromatography analysis.

Figure S5. The overall faradaic efficiency of the CO₂ reduction using Cu/GNF in liquid and gas products has potentials ranging from -0.38 to -0.78 V vs RHE.

As recommended by the reviewer, we have now replotted the Figure 2D Tafel slope using the partial current density of formate production. Furthermore, the electrocatalytic CO₂ reduction activity of the present Cu/GNF was compared with the earlier reports on CO₂ reduction using Cu-based electrocatalysts in Table S3.

Figure 2. D) Tafel plot extracted from the partial current density of the CO₂ saturated LSV.

4. The claimed $R_s = 27$ ohm on page 7 is inconsistent with the Figure 2c, where the ohmic resistances are different between Cu/GNF and GNF.

Response: We thank the reviewer for pointing out this error. It is corrected now in the revised Figure 2C (below).

Figure 2. C) Nyquist plot of GNF with and without CuNPs obtained in 0.1 M KHCO₃ electrolyte at a constant potential of -0.78 V vs RHE within the frequency range from 10 kHz to 0.01 Hz

5. Any references to support the statement of the reduction peaks for copper on Page 6?

Response: As recommended by the reviewer, we have added a new reference (below) to support the statement of the Cu reduction peaks on revised page number 6.

Bard, A. J., Parsons, R. & Jordan, J. *Standard Potentials in Aqueous Solution*. (Routledge, 2017). doi:10.1201/9780203738764

6. The abbreviation for CO₂ reduction is inconsistent (CO₂RR and CO₂R) across the main text. “CO₂” and “H₂O” format need revision on Page 13. Some discussions such as the one on line 218 – 220 are difficult to follow and might need further clarification.

Response: We thank the reviewer for reading our manuscript deeply and pointing out these errors. We revised and maintained the format of CO₂RR throughout the manuscript and corrected the subscript of CO₂ and H₂O.

We revised the sentence highlighted by the reviewer on page number 8 as “Under CO₂ saturation at a potential of -0.38 vs RHE, faradic efficiency for H₂ evolution (FE_{H₂}) represents ≤10% of the overall FE, clearly demonstrating ≥ 90 % selectivity to CO₂RR products”.

Reviewer 3:

1. In the description in line 14, the authors write that the catalyst is deposited in the atomic form while they also report, in line 16, that the catalyst presents 2-5 nm Cu NPs. These two points seem conflicting. Indeed, in the paper, HR-TEM images on fresh catalysts show 2-5 Cu NPs on carbon support without the presence of isolated atoms or without showing the coalescence of single atoms into NPs during sputtering (e.g. with images at different sputtering times). Could the authors make this description clearer?

Response: We thank the reviewer for this comment. We admit that the previous version's sentences might have misled readers about the Cu size. Indeed, the Cu forms 2-5 nm particles. To make it clear, we have revised the main manuscript, including the title (below).

“Direct Deposition of Copper Nanoparticles from atoms at Graphitic Step Edges Lowers Overpotential and Improves Selectivity of Electrocatalytic CO₂ Reduction”.

2. Do the authors have direct evidence of the CO₂⁻ radical as the starting intermediate of CO₂R? The only indirect evidence is the slope in the Tafel plot. However, since it is part of the abstract, I expected that more experiments would be provided to identify this labile intermediate. Furthermore, it occurs at a very reductive potential, which is far from that of the LSV. Can the authors supplement the discussion or provide further evidence?

Response: We thank the reviewer for this constructive comment. We replotted the Tafel slope by extracting the partial current density of formate from the LSV curve. The obtained Tafel slope value of 362 mV dec⁻¹ suggests that the kinetics of the present CO₂ reduction on Cu/GNF catalyst is slow and limited by the diffusion/mass transport of CO₂, which must be the rate-determining step, in agreement with earlier reports.⁴³⁻⁴⁶ This discussion is added in the revised main manuscript on page number 7.

Figure 2. D) Tafel plot extracted from the partial current density of the CO₂ saturated LSV.

3. The last part of the abstract seems to contradict the title. Indeed, from the title, single atoms enhance catalysis by lowering overpotential. However, in lines 23-24, single Cu atoms “disfavour” CO₂RR. Could the authors clarify this point?

Response: We have revised the title of the manuscript as

“Direct Deposition of Copper Nanoparticles from atoms at Graphitic Step Edges Lowers Overpotential and Improves Selectivity of Electrocatalytic CO₂ Reduction”.

In lines 23-24, the text has been revised in the abstract as

“These changes in the catalyst structure make the onset of the CO₂ reduction potential more negative, leading to less formate production at -0.38 V vs RHE”.

4. Line 115, “loss of selectivity”: in the previous lines, the authors discuss that the synthesized carbon/Cu structure favours selectivity of formate at low overpotentials. However, they do not discuss the origin of such loss of selectivity, which is reasonably due to the long-term reaction. Could the authors clarify this point?

Response: We thank the reviewer for this comment. We have added a new text (below) to explain the origin of the loss of activity on page 4 and 5.

“The in-situ catalyst reconstruction over time shifts the CO₂RR onset potential resulting change in selectivity, thus providing a strategy for the future design of highly efficient electrocatalysts for CO₂ reduction”.

5. Lines 158-162: use of RHE and SHE; I suggest using only one reference across the MS.

Response: As recommended by the reviewer, we used RHE throughout the manuscript.

6. Lines 202-205: the sentence is unclear, and it should be rephrased.

Response: As suggested by the reviewer, we have now revised the lines 202-205 on page 8 as below.

“However, as the potential becomes less negative to -0.38 V vs RHE, $FE_{\text{CHOO-}}$ increases sharply to 94% with just 0.84 wt% Cu metal loading”.

7. Figure S5: the total FE is below 80% at large negative voltages. Have the authors possible explanations for the missing FE?

Response: We thank the reviewer for this comment. We have re-examined the data, and provided a revised Figure S5, which shows the overall faradaic efficiency of liquid and gas products close to 100 %.

Figure S5. Faradaic efficiency of detectable gas products from -0.38 to -0.78 V vs RHE

8. Line 238, “CO₂R”: in the text it is mostly occurring as “CO₂RR” could the authors use only one of them?

Response: We thank the reviewer for deep reading of our manuscript and pointing out the typo error. In the revised version, we keep the CO₂RR format throughout the manuscript.

9. Line 261: “(*0.2 mA cm⁻² at -0.38 V vs RHE)” * probably a missing *-* due to a typo. Furthermore, the value disagrees with Figure 3D because at -0.38 V Cu/GNF after 24 h, it shows ca. -0.05 mA cm⁻².

Response: This sentence is revised in page 10 as “LSV analysis of Cu/GNF after 24 hours of the CO₂RR shows a current density of -0.05 mA cm⁻² at -0.38 V vs RHE compared to the initial catalyst, and a negative shift of the onset potential”.

10. Lines 264-267: have the authors checked with HR-TEM the morphology of the best-performing sample after CO₂RR at large negative potentials and compared it with the sample after stability tests? Perhaps this comparison could provide information on the performance loss at a high reaction rate.

Response: We have not checked the morphology changes of Cu/GNF after the reaction at large negative potential using HR-TEM.

11. Lines 294-295: “XPS of fresh catalysts I) and after 24 hours J)” this part of the caption is probably a typo.

Response: We have corrected these typo errors and provided them correctly as “K) XPS spectra of Cu/GNF before the reaction and L) XPS spectra of Cu/GNF after 24 h reaction”.

12. Lines 297-301: there is probably a typo because line 301 mentions the presence of Cu(0), but table S2 shows only Cu(I) and Cu(II) - which is reasonable because Cu(0) and Cu(I) have overlapping XPS signals. A detailed study on Cu XPS signals can be found in DOI: 10.1002/sia.6239. Additionally, the presence of Cu(II) should have two signature satellite peaks at higher binding energies (ca. 934 and 944 eV) with respect to Cu(I)/Cu(0) peaks (at ca. 932 and 941 eV). Could the authors double-check the assignments?

Response: Yes, the referee is right. We corrected the text in the main paper and table S2, following the suggested methodologic paper to differentiate Cu(II) and Cu(0) + Cu (I) contributions for the Cu 2p region.

Revised text in the main manuscript:

“The ratio of Cu species ratio can be determined from the XPS Cu 2p spectra, following the methodology outlined by Biesinger and co-workers.⁵³ This approach assumes an overlap in the contribution of Cu(0) and Cu(I) within the Cu 2p region, making it impossible to differentiate between these two species. After 24 hour reaction, the combined contribution of Cu(0) and Cu(I) increased from 33.7% to 60.4% (**Table S2**), suggesting that the copper species are more reduced after the electrocatalysis”.

13. Lines 318 and 319: CO₂ and H₂O - subscripts are missing.

Response: These subscripts were corrected in the revised manuscript.

REVIEWERS' COMMENTS:

Reviewer #1 (Remarks to the Author):

After thoroughly reviewing the revised manuscript and the authors' responses to the initial reviewer feedback, I see a much better fit and also notice that the work has been improved since the previous submission. Accordingly, I recommend its publication as is. Congratulations to the authors for their commendable contribution.

Reviewer #2 (Remarks to the Author):

Thanks. I have no further comments.

Reviewer #3 (Remarks to the Author):

The authors addressed all my points in the current version of the manuscript. I do not have additional comments and I suggest publication as it is.